# Reporting radiographers' interaction with Artificial Intelligence—How do different forms of AI feedback impact trust and decision switching?

Clare Rainey[1]*, Raymond Bond[2], Jonathan McConnell[3], Ciara Hughes[1], Devinder Kumar[4¤], Sonyia McFadden[1]

1 Ulster University, School of Health Sciences, York St, Belfast, Northern Ireland, 2 Ulster University, School of Computing, York St, Belfast, Northern Ireland, 3 NHS Leeds Teaching Hospitals, Leeds, United Kingdom, 4 School of Medicine, Stanford University, California, United States of America

¤ Current address: Head–MLOps, Layer6 AI
* c.rainey@ulster.ac.uk

**Data Availability Statement:** Data cannot be shared publicly because of ethical restrictions on sharing sensitive data. Access to the data can be

## Abstract

Artificial Intelligence (AI) has been increasingly integrated into healthcare settings, including the radiology department to aid radiographic image interpretation, including reporting by radiographers. Trust has been cited as a barrier to effective clinical implementation of AI. Appropriating trust will be important in the future with AI to ensure the ethical use of these systems for the benefit of the patient, clinician and health services. Means of explainable AI, such as heatmaps have been proposed to increase AI transparency and trust by elucidating which parts of image the AI 'focussed on' when making its decision. The aim of this novel study was to quantify the impact of different forms of AI feedback on the expert clinicians' trust. Whilst this study was conducted in the UK, it has potential international application and impact for AI interface design, either globally or in countries with similar cultural and/or economic status to the UK. A convolutional neural network was built for this study; trained, validated and tested on a publicly available dataset of *MU*sculoskeletal *RA*diographs (MURA), with binary diagnoses and Gradient Class Activation Maps (GradCAM) as outputs. Reporting radiographers (n = 12) were recruited to this study from all four regions of the UK. Qualtrics was used to present each participant with a total of 18 complete examinations from the MURA test dataset (each examination contained more than one radiographic image). Participants were presented with the images first, images with heatmaps next and finally an AI binary diagnosis in a sequential order. Perception of trust in the AI systems was obtained following the presentation of each heatmap and binary feedback. The participants were asked to indicate whether they would change their mind (or decision switch) in response to the AI feedback. Participants disagreed with the AI heatmaps for the abnormal examinations 45.8% of the time and agreed with binary feedback on 86.7% of examinations (26/30 presentations).'Only two participants indicated that they would decision switch in response to all AI feedback (GradCAM and binary) (0.7%, n = 2) across all datasets. 22.2% (n = 32) of participants agreed with the localisation of pathology on the heatmap. The level of agreement with the GradCAM and binary diagnosis was found to be correlated with trust

provided following a successful application to Ulster University's Nursing and Health Research Ethics Filter Committee. Ulster University's Research Portal contains metadata on the dataset and instructions on how to request access to this dataset. This information can be accessed at https://doi.org/10.21251/50890091-4b54-4644-b980-ea9da646aa0e.

**Funding:** This work has been funded by the College of Radiographers Research Industry Partnership Research awards scheme (CoRIPS) no. 183. The funder had no role in study design, data collection and analysis, decision to publish, or preparation of the manuscript.

**Competing interests:** The authors have declared that no competing interests exist.

(GradCAM:—.515;—.584, significant large negative correlation at 0.01 level (p = < .01 and —.309;—.369, significant medium negative correlation at .01 level (p = < .01) for GradCAM and binary diagnosis respectively). This study shows that the extent of agreement with both AI binary diagnosis and heatmap is correlated with trust in AI for the participants in this study, where greater agreement with the form of AI feedback is associated with greater trust in AI, in particular in the heatmap form of AI feedback. Forms of explainable AI should be developed with cognisance of the need for precision and accuracy in localisation to promote appropriate trust in clinical end users.

## Author summary

Artificial Intelligence (AI) is pervading many areas of our day-to-day lives, including healthcare. In radiology, AI models are being used to assist in the interpretation of radiological images. Professional communities are aware of the impressive performance of these systems but less aware of the relationship between AI and the human end-user. The expert/experienced users' perception of trust in these systems has been cited as a major barrier to the successful integration of healthcare AI. This study uses 'reporting radiographers', who provide written diagnoses on plain radiographic images, to investigate the impact of the type of AI feedback offered on the expert users' trust perception. Participants were presented with two commonly used forms of AI feedback–a binary diagnosis, with AI confidence indication, and a visual form of feedback ('heatmap') to indicate the area of the image the AI found most important in reaching its decision. The study found that agreement with the AI feedback is an important indicator of trust, particularly with the 'heatmap'. This is important in the design of the user-interface in clinical AI, where acceptance of the form of feedback may prove to be a barrier or a facilitator to the integration of AI into clinical practice.

## 1 Introduction

Studies investigating the potential of 'Artificial Intelligence' (AI) as a human adjunct in detecting pathology on radiographic images began in the 1960s with a 'proof of concept' system converting images to numerical data [1]. Since then, electronic evolutionary changes with rapidly increasing computational power availability has permitted the development of ever more sophisticated detection systems. Differing methods of analysis of medical, and other, images have been developed, with Deep Learning (DL) using Convolutional Neural Networks (CNN) being the most recent and seemingly most promising form of AI for detecting disease from radiographic images. CNNs are trained to perform certain specific tasks and develop accuracy based on feedback. The algorithm therefore develops its performance ability rather like a human brain with layers of 'neural networks' built up to allow improvement with experience. This is referred to as supervised Machine Learning (ML) where a large set of pre-labelled cases are used to 'teach' the computer to discriminate between normal and abnormal cases.

These developments have been the focus of much media and professional attention. The development of AI has been targeted as an area of focus for modernising and future-proofing the National Health Service in the UK [2]. There are several significant drivers to the development of AI as a tool in the health care setting; namely, time constraints / efficiency, error avoidance or minimisation and workflow augmentation [3,4]. It is estimated that the implementation of an effective AI system for automated image reporting could reduce the time that

radiologists spend reviewing images by 20% [5], and thus liberate 890,000 hours of radiologist time per annum [5]. This time can be spent doing non automatable tasks such as providing personalised patient care and more complex tasks where human input is essential. This is particularly important in worldwide healthcare systems, coping with current multi-faceted pressures, including the exacerbation of these resulting from the COVID-19 pandemic, where resources are limited [6].

Despite reported accuracies of AI for a number of applications in radiology [7,8,9], clinicians' trust in AI remains a barrier to implementation in the health care setting [10]. This is particularly the case with the use of DL. As mentioned, DL algorithms make use of multiple neural layers to analyse and process image data but there are a number of these layers which are hidden to the user. It is not apparent, therefore, how the algorithm reaches its ultimate decision. This, not only, has implications for users' trust in these systems but should be central when considering the ethical issues surrounding the clinical use of AI in radiology [11]. Attempts are currently being made to make this process more transparent by the use of visual representations of the areas of attention of the AI, for example, heat or attention maps [12,13,14].

Plain radiographs of the appendicular skeleton are commonly performed examinations, with 23.2 million requested in NHS England per annum [15], therefore clinicians are accustomed to using them in the clinical setting. Despite plain radiography being commonly used in clinical practice, the use of AI in identification of abnormality on appendicular skeletal radiographs remains a relatively unexplored area, with the first publication of experimental results in 2017 [16]. Other studies followed, reporting impressive accuracies [17].

This proposed study aims to use a larger dataset of projections for seven anatomical areas, that includes fingers, hand, wrist, elbow, forearm, shoulder and humerus, in order to better replicate the range of patient presentation in the clinical setting.

Radiographer reporting is an established practice in the UK, however the NHS Diagnostics: Recovery and Renewal report (2020) [18] calls for this role to be further expanded to allow 50% of all plain radiographic images to be reported on by a radiographer. Whilst this is a welcome recommendation, the current dearth of radiographers and radiologists in the UK may mean that, not only must the number of radiology professionals (radiographers and radiologists) being trained increase, but efficiencies must be found, for example, the integration of AI to assist with faster reporting turnaround times [18,19,20,21] as suggested by NHS England draft document of February 2023 titled 'Diagnostic Imaging Reporting Turnaround Times [22]

This study attempts to quantify expert radiographers' trust in AI during and after exposure to AI binary diagnoses and heat maps on plain radiographs of the appendicular skeleton.

## 2 Aim and objectives

This study's principal aim was to investigate the factors impacting reporting radiographers trust in Artificial Intelligence (AI) for use in image interpretation. Musculoskeletal radiographs of the upper extremity from the MURA dataset [23] were used.

The objectives of this study were to:

i.  investigate reporting radiographers' trust in an AI system used to provide clinical decision support on radiographic images of the upper appendicular skeleton.

ii.  investigate the factors impacting trust in these systems, including agreement with heat-maps, used as means of explainable AI and binary diagnosis from the AI.

iii.  determine the propensity of the users to decision switch following provision of the AI feedback

## 3 Materials and methods

### 3.1 Ethical permission

Ethical permission was granted from Ulster University Nursing and Health Research Filter Committee (FCNUR-20-026).

### 3.2 Model training

The MURA dataset is a publicly available dataset used for training AI models for computer vision tasks in pathology detection on plain radiographs of the upper appendicular skeleton. This ran as a 'competition' where competitors submit their code to a central repository (https://codalab.org/) for testing on an unseen test set. The dataset consists of 40,561 radiographic images (14,863 examinations, training dataset n = 36,808 images) which have ground truth established (normal/abnormal). The dataset is split into training and validation datasets. The training set was used for initial training of the model and the validation set was used to fine-tune the parameters until an acceptable accuracy was reached.

The images in the dataset vary in quality, resolution, aspect ratio. The images are exclusive to each dataset, with no overlap.

### 3.3 Test dataset

The test set of the MURA competition is not publicly available. For this study half the validation set was used as a test set, consisting of 783 patients, 1,199 studies and 3197 images. The remainder was used as a 'validation set'. The MURA competition has now closed and, therefore, there were no established diagnoses available for the testing and validation sets, however, as the participants were practicing reporting radiographers, their consensus diagnosis was taken as 'ground truth'.

No overlap between any of the sets was permitted. The test set was chosen to contain approximately half of each of the seven upper extremities for adequate and balanced representation of each class.

### 3.4 AI model

A pretrained (ImageNet) ResNet-152 convolutional neural network (CNN) was used in this study. For each examination, the arithmetic mean of the output is used to determine pathology, with any probability over 0.5 deemed abnormal. The model was trained on the training set until no further improvement was observed. Adam optimiser was used with initial learning rate to $10^{-4}$.

### 3.5 Salience map

A saliency map was produced for each image to 'explain' the AI prediction. This was created using a technique used by one of the authors in a previous study [12]. The whitest area of the colour map indicated the strongest areas of spatial location used by the model to determine the probability. Null values are represented as black.

### 3.6 Patient-public involvement

A patient-public involvement group was established to advise on the relevance of this study for key stakeholders. Feedback was also sought on the methodology of the study and plans for dissemination, including the appropriateness of presentation at conferences and publishing the work in international, peer reviewed journals [24]. The group consisted of patients and

radiographers (n = 5). The background, rationale and expected impact of this study was discussed over one face to face meeting and subsequent email updates.

### 3.7 Pilot study

The study was piloted amongst three experienced radiographers, two from a clinical reporting background and one from an academic institution, with extensive experience in plain radiography. Each of the pilot participants had in excess of 20 years clinical and/or academic experience.

A random sample of radiographs not included in the main study were selected and embedded into the Qualtrics platform. Participants were asked to provide feedback on the suitability of the questions asked, the accessibility of survey platform, time taken for completion and the quality of the radiographic images, thus ensuring both face and content validity [25]. As a result of this, the number of images which would be included in the main study was decided and based on an approximation of the number of images that could be interpreted in a maximum of 90 minutes. Participants in the pilot study reported that they found the study easy to follow and comprehend. No changes were made to the study flow in Qualtrics as a result of the pilot study, however, image quality was noted as an important issue. Participants reported that many of the images in the dataset were of low diagnostic quality to the human eye. A question relating to the participants' perception of the image quality of each presented image was then included in the final version of the study to ensure all examinations included in the datasets were acceptable for interpretation by the participants.

### 3.8 Recruitment / Participant selection

A request for reporting radiographers to participate in this study was made through authors' professional networks including reporting and advanced practice special interest groups (The Society of Radiographers (SoR) Consultant Radiographers Advisory Group and The SoR Reporting Radiographers Special Interest group). Interested participants were directed how to contact the researcher (CR) by email. The first three interested participants in each region of the UK (England, Scotland, Wales and Northern Ireland) were recruited to the study. A £25 gift voucher was offered following expression of interest to take part in the study to encourage completion of the study, therefore minimising the risk of attrition. All participants completed the entire study to completion.

Each participant was emailed a unique link to the survey on the 18[th] July 2022 and a reminder email sent on the 2[nd] August 2022, resulting in full participation. A unique link was provided to ensure individuality of responses to confirm the resultant data met the requirements to allow for analysis of interrater agreement using Fleiss kappa [26].

A power calculation was not carried out as this is a novel study with lack of comparable quantitative research in the area. The literature indicates that in these cases a sample size of 10–15 participants is adequate for this type of study [27,28].

### 3.9 Construction of the dataset for Qualtrics

Each imaging series in each case of the MURA test set were initially analysed by one of the authors not involved in the interpretation (CR) and were excluded from this study if the image contained surgical hardware, for example, following internal surgical fixation of trauma or any artefact felt to be significantly obscuring anatomy as demonstrated in Fig 1. The overall diagnostic quality of the images included in the main study were then reviewed by the remining authors with experience in radiographic image interpretation (SMF and JMcC) and rejected as necessary (Figs 2 and 3). The rationale of each exclusion was tabulated and is available, upon request, from the corresponding author (CR). Following exclusions, 299 imaging series were

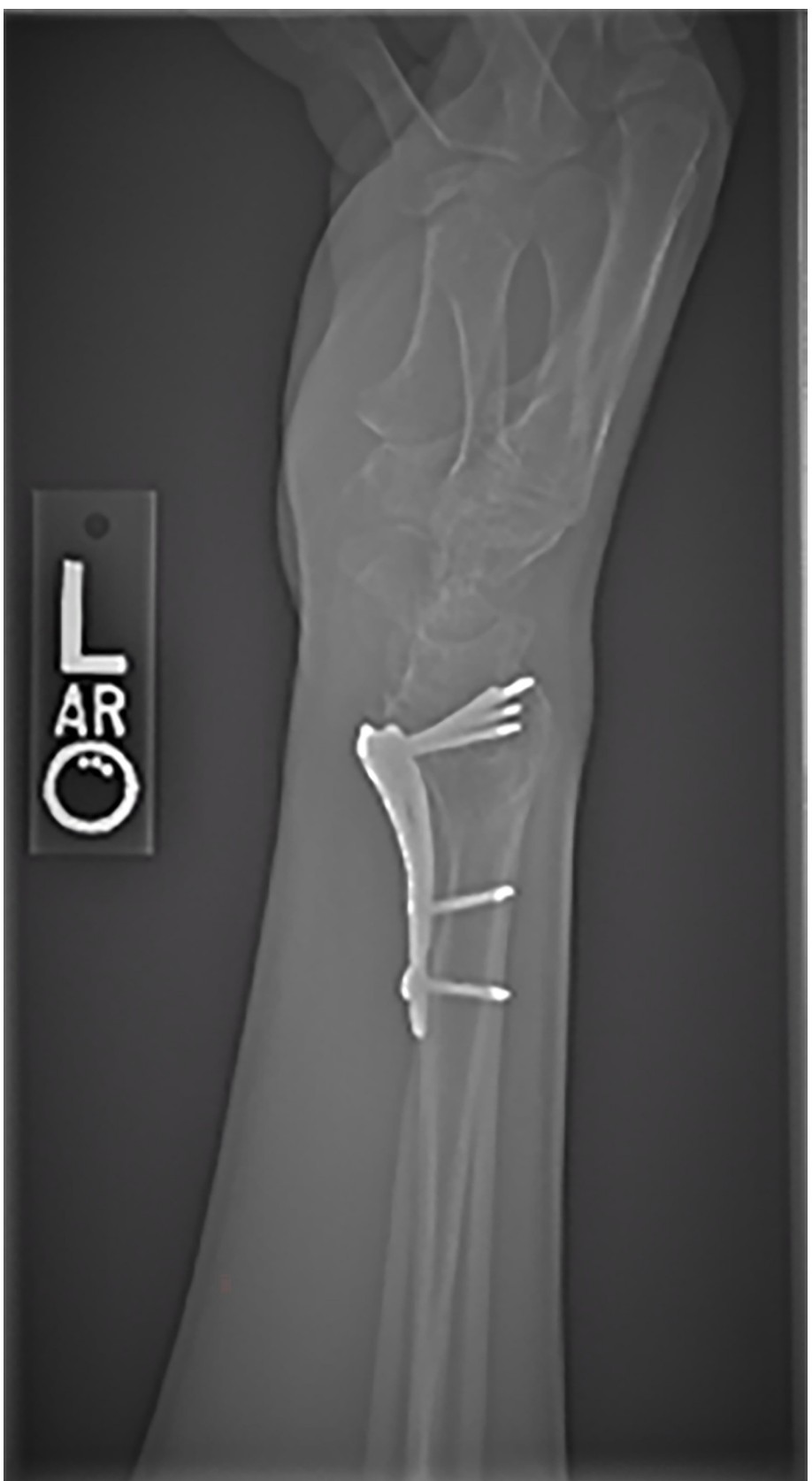

**Fig 1. Example of rejected image due to presence of internal fixation.**

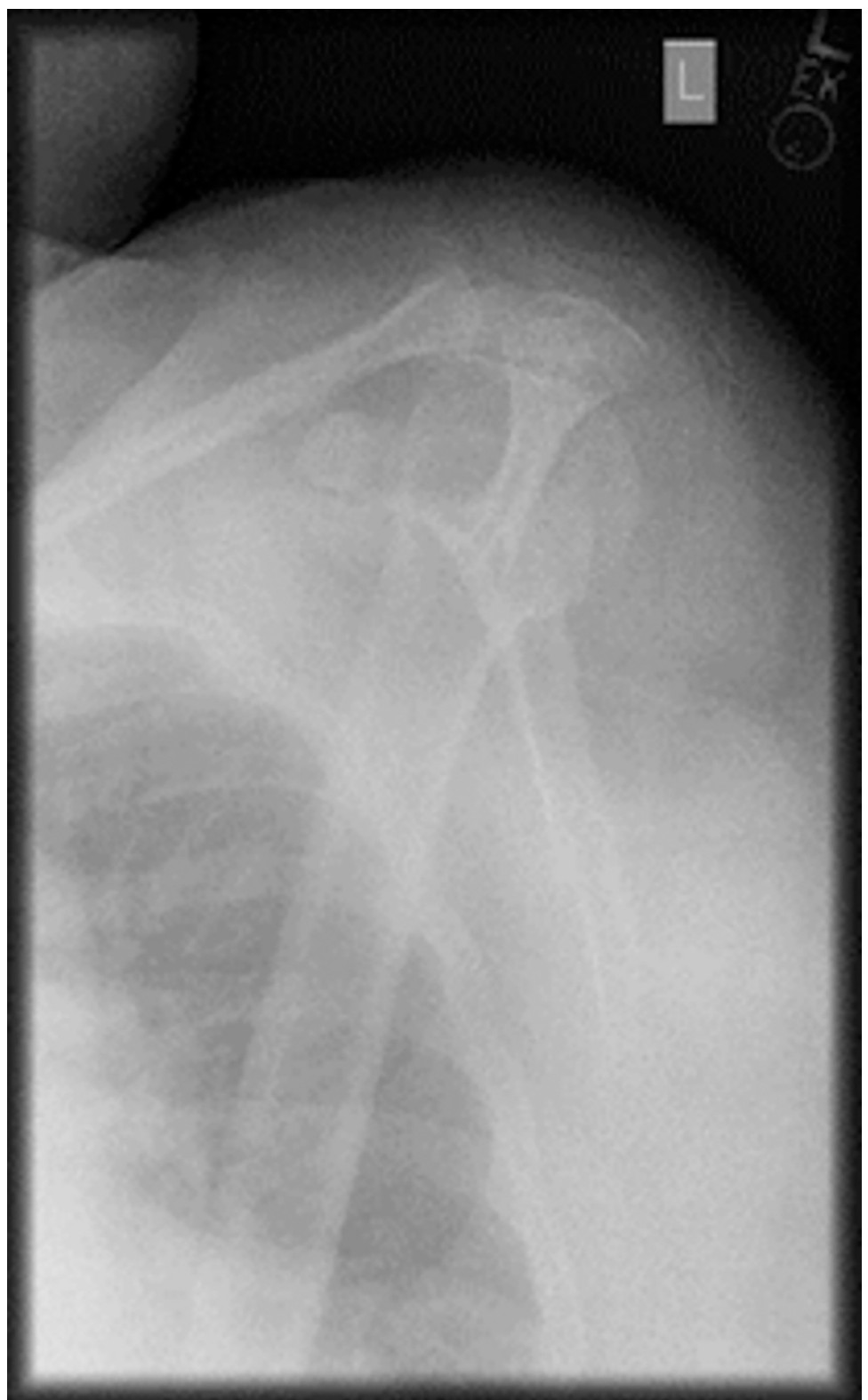

**Fig 2. Example of rejected image due to poor image quality.**

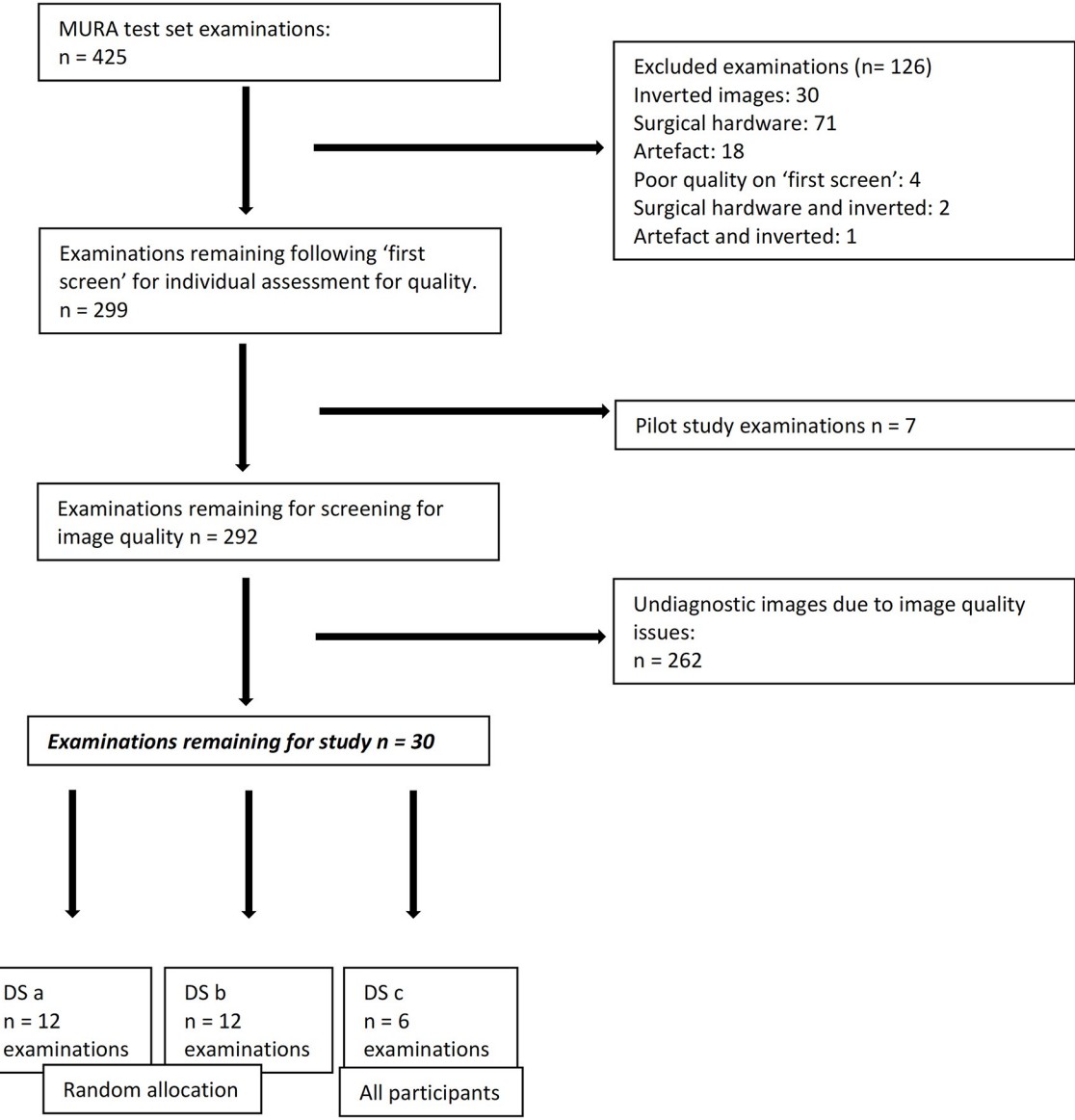

**Fig 3.** dataset building: Dataset (DS) a, b and c.

included in this study. Each imaging series contains one or more projections of the anatomical area. Each imaging series was allocated a number from 1–299. A random number generator (Random.org) was used to select 110 examinations. Each examination was then closely inspected by the author (CR) for diagnostic acceptability and verified by another two authors involved in the pilot study (SMF, JMcC).

The remaining 30 full examinations (minimum two images in each examination) were divided into two datasets of 12, and one dataset of six examinations. Each participant was randomly allocated one of the datasets of 12 examinations and all participants were presented with the dataset of six examinations, to allow the author to compute the interrater reliability across all participants (S1 File).

Each image was adjusted to optimise viewing by modifying light, clarity and colour in Microsoft Photos and saved as a.png image. This is a lossless compression format and has been

used in a similar study [14]. Three authors independently reviewed each image for quality assurance purposes (CR, SMF, JMcC).

The radiographic images were randomly assigned to each dataset and embedded in the Qualtrics survey platform. Each dataset (DS) was assigned a letter (a, b or c) and each examination a number (1–12 for DS a and b or 1–6 for DS c), for example, DS a: examination 1, DS a, examination 2, and so on. Each participant was randomly allocated one of dataset (DS) a or b using the randomiser function in Qualtrics. Each participant was allocated DS c. In total, each participant was asked to interpret 18 radiographic examinations.

Each of the participants taking part in the study received their own individual link to the study. Internet Protocol (I.P.) addresses were not gathered to ensure full anonymity. Participants were provided with an information and instruction sheet via email. Included in this were contact details of the Research Governance team at the institution. The purpose for the data collection, means of analysis and dissemination strategy was provided. Consent was indicated and recorded by the participants on an initial slide in the Qualtrics platform used for this study. If participants indicated that they did not consent to taking part, they were exited from the survey and no data was collected. No participant indicated that they did not consent.

Participants were asked to indicate their binary diagnosis and localisation of any pathology on each image, before and after provision of a Gradient-weighted Class Activation Map (GradCAM) 'heatmap' overlay. They were then asked to indicate the extent of the heatmap agreement with their localisation of pathology (if applicable). They were asked to indicate their trust in the system following provision of the heatmap. All the images in the examination were presented again to the participant, along with the binary diagnosis (pathology/no pathology) from the AI. Participants were asked to indicate their level of trust in the AI system following both heatmap and binary feedback. They were then asked if the AI feedback would have caused them to change their mind from their initial decision. Finally, they were asked to rate the diagnostic quality of the images included on a scale of zero to five (S1 File).

### 3.10 Data analysis

Interrater agreement across all readers on all datasets was established using Fleiss' kappa to determine the level of agreement across two or more raters [26].

Descriptive statistics describe the agreement of the AI and the user, using consensus diagnosis of the participants as ground truth. Consensus was determined as the most popular decision of the participants (S2 File). Percentage agreement with the AI diagnosis was established.

Ordinal and nominal perception data was obtained to gain an insight into the users' trust, therefore non-parametric tests were used for correlation. Spearman's rho ($r_s$) and Kendall's kau ($\tau_b$) were used to investigate any correlation between:

1. Trust (0–5) and level of agreement with the heatmap feedback from the AI (perfect agreement (coded 1), partial agreement (2) and disagreement (3))

2. Trust (0–5) and level of agreement with the binary feedback from the AI (perfect agreement (1), partial agreement (2) and disagreement (3)

3. Trust (0–5) and propensity of the user to change their mind from their initial decision (yes (1), maybe (2), no (3)

4. Image quality (0–5) and agreement with binary AI feedback (perfect agreement (1), partial agreement (2) and disagreement (3)

5. Image quality (0–5) and trust (0–5)
Image quality (0 representing no trust– 5 representing absolute trust) and propensity of the user to change their mind from their initial decision (Yes (1), maybe (2), no (3))

6. Level of agreement with AI was possible only on images where abnormality was detected. Data entries indicating that there was no abnormality, as determined by the consensus diagnosis of the participants, were coded as 'missing data' before analysis.

The strength of any relationship was determined using the suggestion by Cohen (1988), cited in Pallant, 2007 [29]:

Small .01 to .29
Medium .30 to .49
Large .50 to 1.0

## 4 Results

### 4.1 Participants

There were 12 participants in this study, three from each region of the UK, namely, England, Scotland, Wales and Northern Ireland. Each participant was a practicing reporting radiographer, a registered radiographer who has received extra postgraduate academic and clinical training to provide written diagnoses on radiographic images.

### 4.2 Interrater agreement

Interrater agreement was established on all examinations within each of the datasets. Fleiss' kappa (k) was calculated and indicated:

moderate agreement between participants allocated dataset (DS) a (n = 5, k = .581, 95% C.I. .475 - .688, p < .05),

moderate agreement between participants allocated DS b (n = 7, k = .410, 95% C.I. .338 - .482, p < .05), and fair agreement between participants allocated DS c (n = 12, k = .228, 95% C. I. .172 - .283, p < .05). Agreement was calculated for individual categories (abnormal and normal). The magnitude of agreement was determined by levels determined for the Cohen's kappa coefficient, as suggested by Altman (1999 [30]):

Poor < .02
Fair .21 to .40
Moderate .41 to .60
Good .61 to .80
Very good .81 to 1.00

No statistically significant difference in responses was found for any dataset, concluding that participants responded in a similar way, therefore descriptive statistics below are reported per image and per examination, rather than per participant.

### 4.3 Agreement of participants with AI heatmap

Following presentation of the heatmap for each of the images in each examination the participant was asked to indicate their level of agreement with the heatmap localisation. This was only relevant when the participant deemed there was abnormality present on the image. Level of agreement was measured by asking the participant to choose from one of the following options:

• Yes, in all areas I have previously identified

- Yes, in more than half of the areas I have previously identified

- Yes, in less than half of the areas I have previously identified

- No, there are no areas of agreement with areas I have previously identified

- No pathology

    These findings were coded into four categories for analysis:

- Perfect agreement (Yes, in all areas I have previously identified, coded '1')

- Partial agreement (yes, in more than half and less than half, coded '2')

- Disagreement (no, there are no areas of agreement, coded '3') and

- No pathology.

The responses indicating there was no pathology were coded as missing data and excluded from the analysis.

Following removal of the 'no pathology' responses, there were 144 decision points remaining. There was perfect agreement with the heatmap noted in 22.2% of images (n = 32), partial agreement in 31.9% of images (n = 46) and disagreement in 45.8% of images (n = 66). Data is presented in full in Figs 4 and 5.

### 4.4 Agreement of participants with binary AI diagnosis

Initially, agreement of the participants with the AI was established by consensus diagnosis for each examination (S2 File). The participants agreed with the binary AI feedback on 86.7% of the examinations (n = 26 out of 30 examinations). Of those where the consensus diagnosis of the participants differed from the AI binary feedback, three were false positive, where the AI determined pathology in examinations where the radiographer did not and one false negative, where the AI failed to detect a pathology identified by the participant (Fig 6). Image quality was rated below the mean image quality (3.1) of the study for all these examinations (2.6, 3.0, 2.1 and 2.0 for Patients 3a, 5a, 3b and 5b respectively).

Further analysis per participant for each examination indicated that there was agreement with the AI, either perfect or partially, for all examinations except for examination 3b, where all participants disagreed with the AI decision (100%, n = 7 participants) and examination 5b, where there was disagreement with the AI indicated by most of the participants (86%, n = 6 participants) (Fig 6)).

Overall, there was agreement with the AI binary diagnoses in 66.9% of instances (i.e., n = 147 out of a total of 216 decision points (12 participants' decisions over 30 examinations)), partial agreement 22.5% (n = 46) and no agreement in 11.2% of examinations (n = 23). Full detail is presented in Figs 7 and 8, including indication of the instances of disagreement with the AI decision.

### 4.5 Decision switching

Decision switching is the self-reported likelihood of the participants to change their mind from their initial diagnosis. Detailed findings, per dataset, are presented in Figs 9 and 10. There were 60 decision points in response to this question for each dataset (n = 5 participants for DSa (12 examinations), 7 for DSb (12 examinations) and 12 for DSc (6 examinations)), resulting in a total of 216 decision points overall. Of the total 216 responses to this question across all datasets there were only two instances (examination 11b and 1c) where participants indicated that they would change their mind in response to the AI feedback (1.1%), n = 35

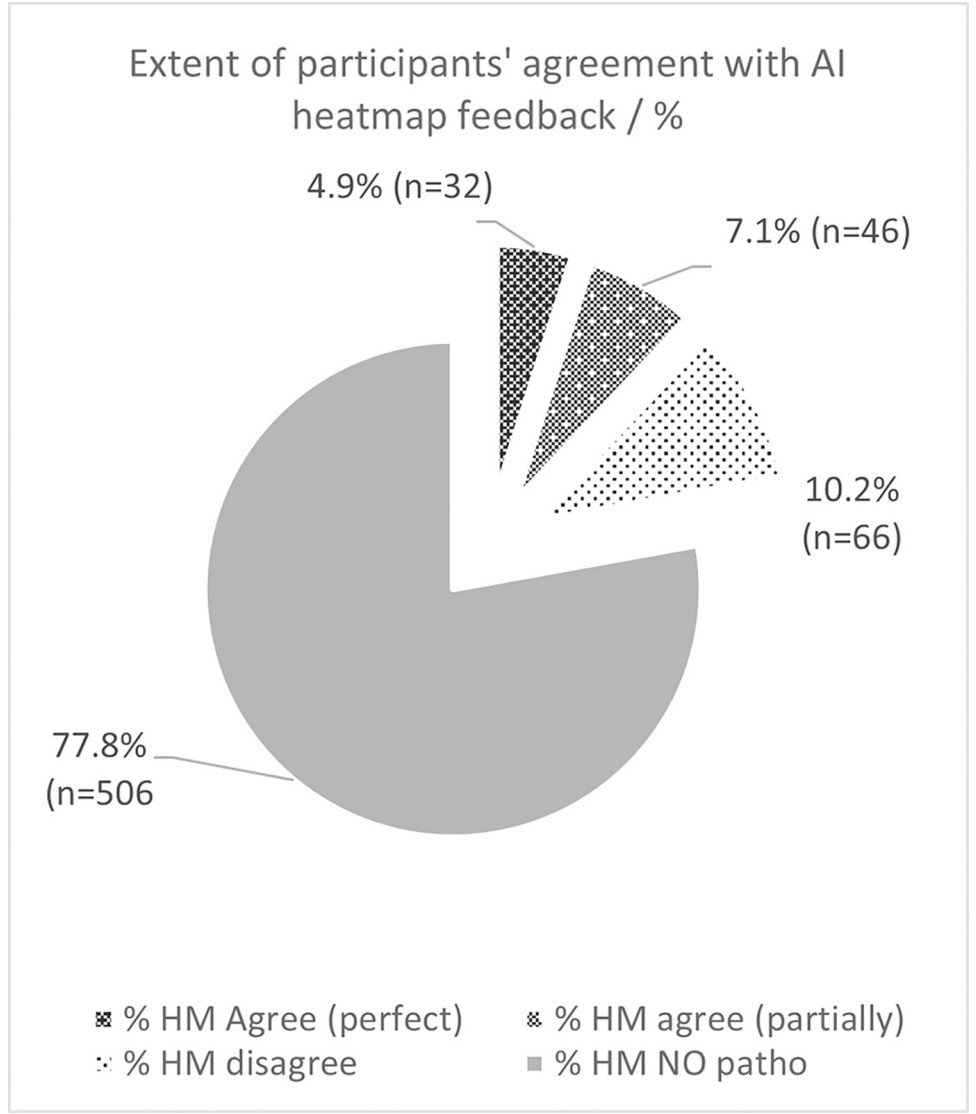

**Fig 4. Participant agreement with heatmap feedback.**

(15.7%) indicated that they might change their mind from their initial diagnosis and the remainder (n = 179, 83.5%) indicated that they would retain their initial decision.

## 4.6 Image quality

In response to the feedback to the pilot study regarding poor diagnostic quality of some of the images, participants were asked to rate the quality of the images in each examination with 0 indicating an undiagnostic image and 5, perfect image quality. The mean score across all images in this study was 3.1.

## 4.7 Correlation analysis (Table 1)

Users' trust perception in AI was obtained after the heatmap presentation for each radiographic image and after the presentation of all images (including heatmaps and binary feedback) in each examination (S1 File).

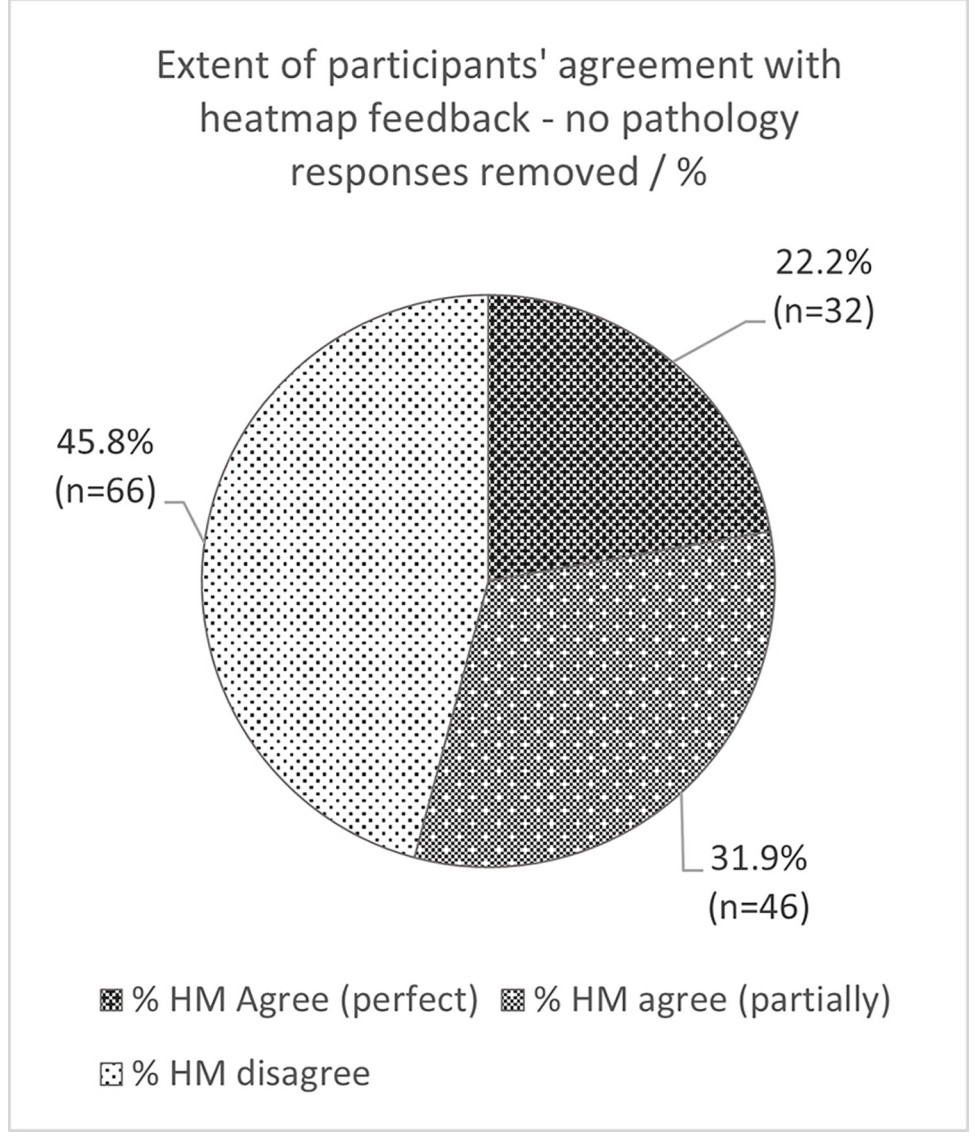

**Fig 5. Participant agreement with heatmap feedback (pathological images only).**

There was a statistically significant large negative correlation between users' trust, per image (0–5) and the level of agreement with the heatmap for each image (perfect, partial and no agreement) (n = 645, $\tau_b$ = —.515, $r_s$ = —.584, p = < .01)

There was also a statistically significant moderate negative correlation between trust, obtained at the end of the examination (following all images, heatmaps and binary AI feedback) and level of agreement with the binary feedback from the AI (perfect agreement, partial agreement and disagreement) (n = 216, $\tau_b$ = —.309, $r_s$ = —.369, p = < .01)

A statistically significant small negative correlation was also found between the users' perception of image quality and agreement with binary AI feedback at the end of the examination (n = 126, $\tau_b$ = —.238, $r_s$ = —.268, p = < .01). Finally, a statistically significant small positive correlation was found between image quality and trust (n = 216, $\tau_b$ = .219, $r_s$ = .256, p = < .01)

*Patient 3a All images from this examination are given again below. The AI system determined that this examination/imaging series DID NOT contain evidence of pathology. Do your initial diagnoses agree with this?*

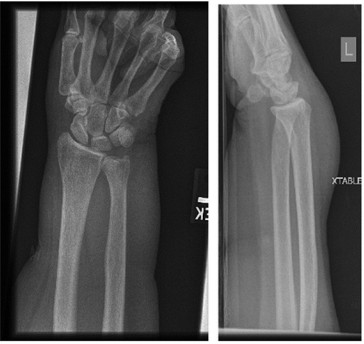

*Patient 5a All images from this examination are given again below. The AI system determined that this examination/imaging series DID contain evidence of pathology. Do your initial diagnoses agree with this?*

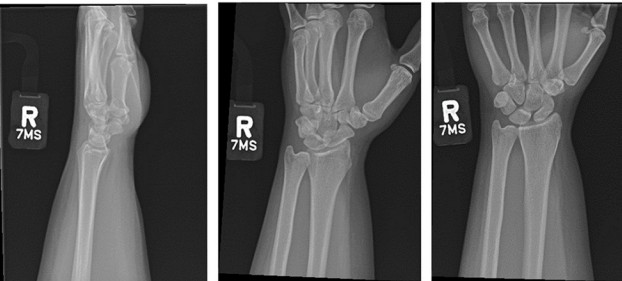

*Patient 5b: All images from this examination are given again below. The AI system determined that this examination/imaging series DID contain evidence of pathology. Do your initial diagnoses agree with this?*

*Patient 3b: All images from this examination are given again below. The AI system determined that this examination/imaging series DID contain evidence of pathology. Do your initial diagnoses agree with this?*

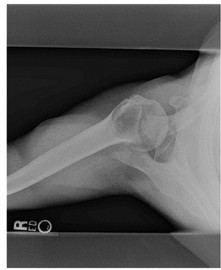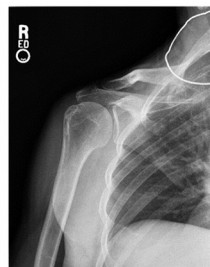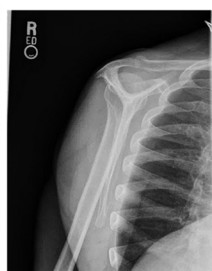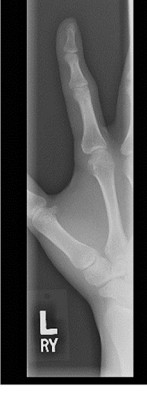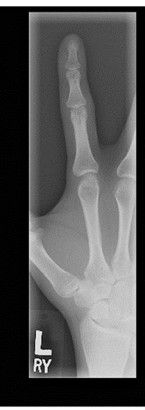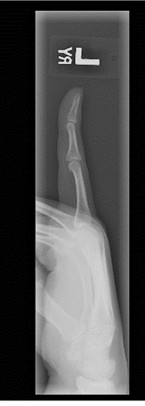

**Fig 6. Instances of human/AI diagnostic disagreement with AI Binary feedback.**

There were no statistically significant correlations between trust and propensity to change their mind from the initial decision, or between the quality of images and propensity to change their mind (Full detail is provided in Table 1).

In summary, as agreement with the heatmap and binary AI feedback increases, trust increases. With this group of participants, the decision switching rate does not seem to be correlated with trust in the AI. As the quality of image increases, the users' agreement with AI feedback and trust perception increases. There is no correlation found between image quality and decision switching rate with this participant cohort.

## 5 Discussion

### 5.1 Participants and interrater agreement

There was fair to moderate agreement across all participants in this study, in all included examinations (n = 30). All participants were allocated DS c and therefore this may have contributed to the increased variability in responses.

### 5.2 Agreement of participants with AI heatmap

Methods of explainability have been suggested as a way to calibrate trust and ensure responsible use of AI systems, i.e., allowing the user to interrogate the system to make a decision on what their trust level should be on a case-by-case basis, rather than in AI technologies as a

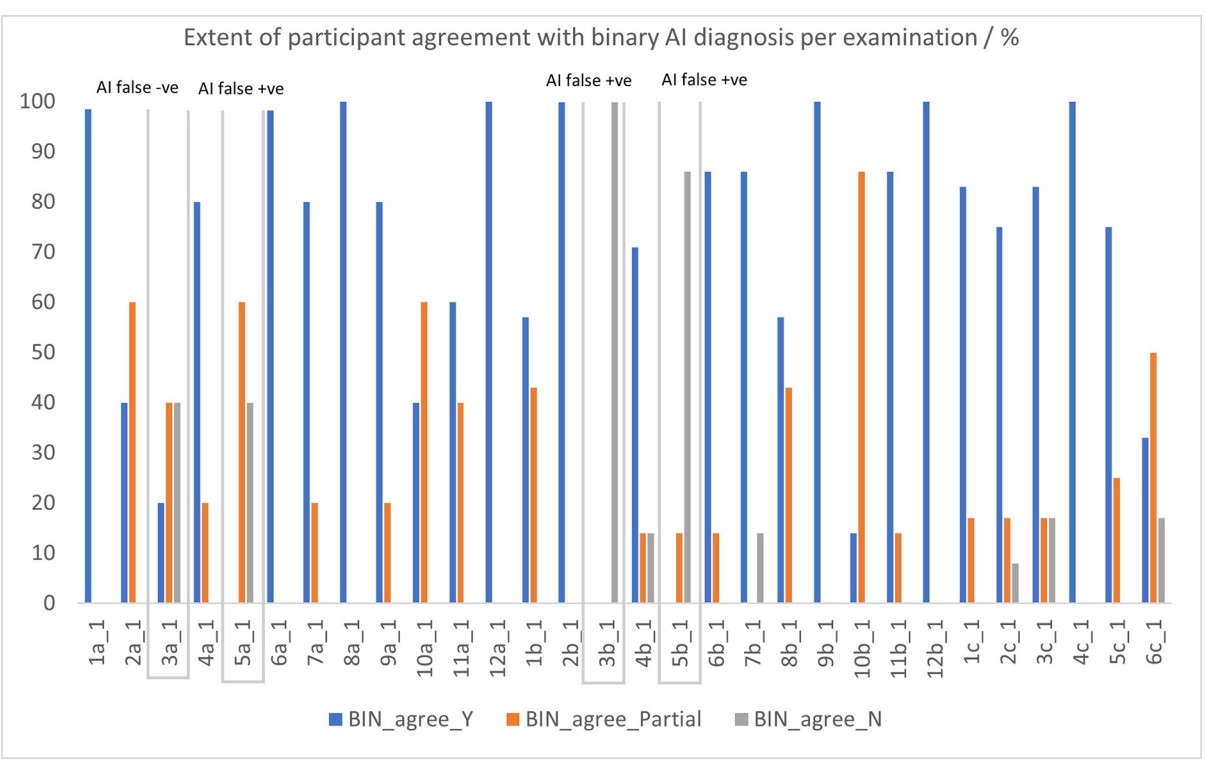

**Fig 7. Participant agreement with binary AI diagnosis per examination.**

whole. Studies have suggested that heatmaps may increase trust and contribute to the ethical use of AI [11]. This may be of particular importance when DL AI systems are being integrated into the clinical setting to be used in clinical decision support, where there is much discussion regarding the ethical implications of the use of AI in health care mooted in the professional

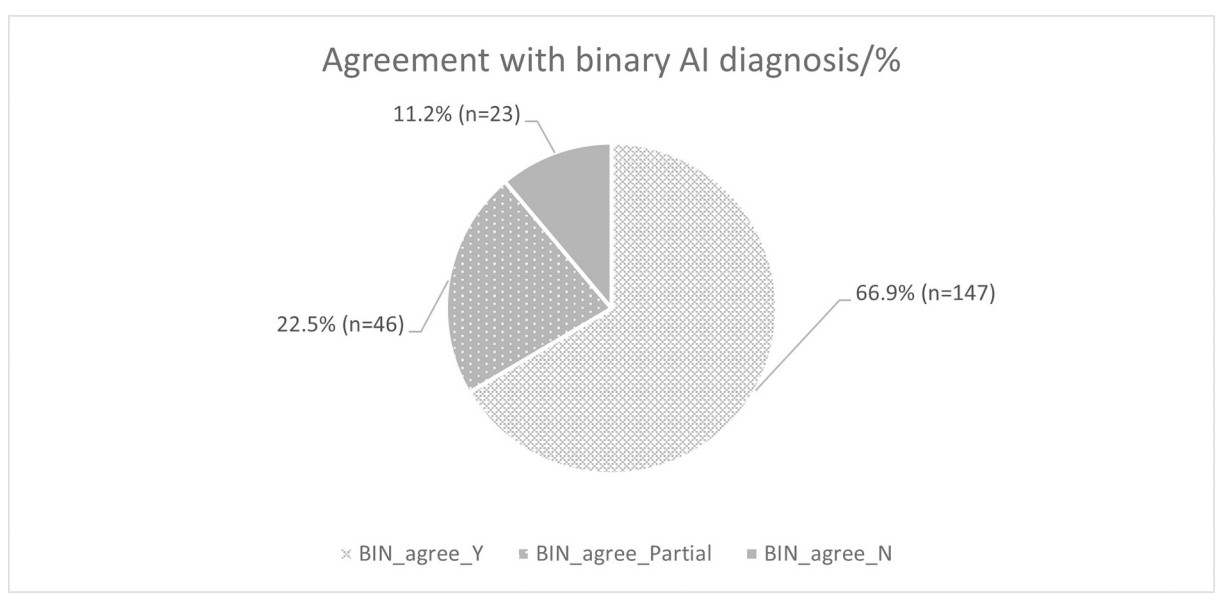

**Fig 8. Participant agreement with binary AI diagnosis for all datasets.**

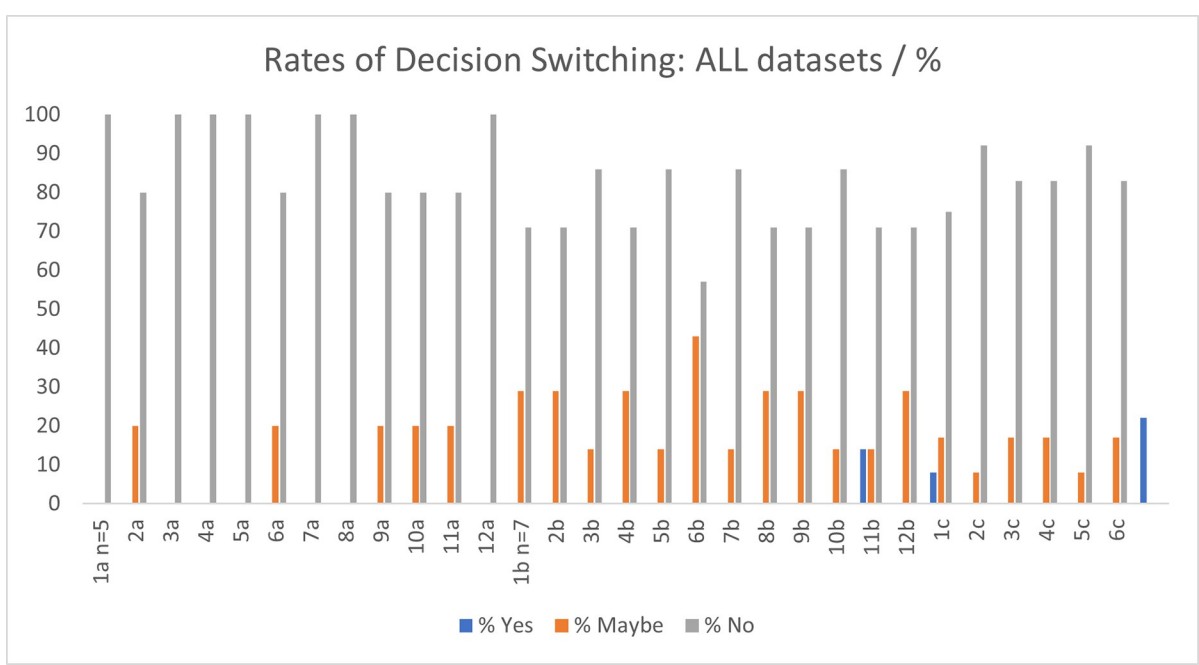

**Fig 9. Decision Switching: Participants' perception of their likelihood to change their mind from their initial diagnosis following** *heatmap* **feedback from the AI model.**

literature [31,32]. There have been a number of methods of AI explainability proposed, with heatmaps indicating the area of focus of the AI in making its decision a popular choice [12,33].

As expert interpreters of radiographs the participants in this study disagreed with the heatmaps almost half the time (45.8%, n = 66). They indicated perfect agreement with the heatmap on 22.2% of occasions. Previous research has found that the visual feedback form of AI

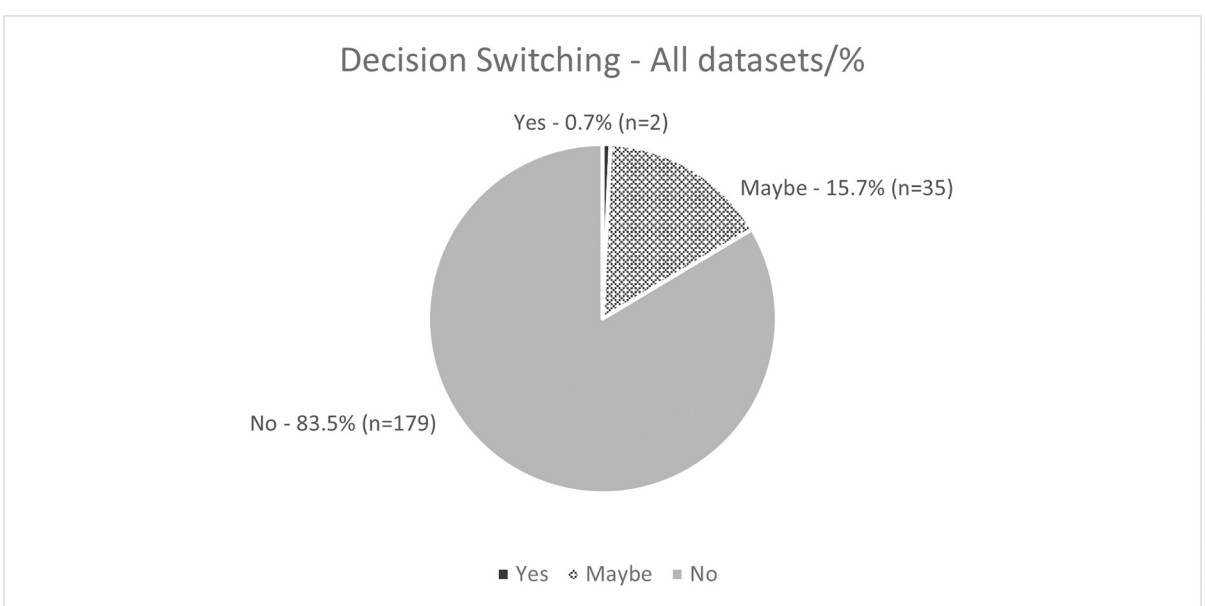

**Fig 10. Decision Switching: Participants' perception of their likelihood to change their mind from their initial diagnosis following** *binary* **feedback from the AI model.**

**Table 1. Correlation analysis.**

| | INDEPENDENT | DEPENDENT | Correlation<br>*(Kendall's tau and Spearman's rank respectively)*<br>*Effect size: small .01, medium .30, and large .5 (Cohen, 1988)* |
|---|---|---|---|
| **Correlation between agreement with *image heatmap* and *trust*** <br>*n = 645 decision points* | Agreement with HM (perfect (1), partial (2), disagreement (3)) n = 645 (decision points) | Trust perception (0–5, 0 representing *no trust* and 5 *absolute trust*) n = 645 (decision points) | - .515;—.584, significant large negative correlation at 0.01 level (p = < .01) |
| **Correlation between agreement with *binary AI feedback* (HM and binary) and *trust*** <br>*n = 216 decision points* | Agreement with binary feedback (yes (1), partly (2), no (3)) | Trust perception (0–5, 0 representing *no trust* and 5 *absolute trust*) following *all* AI feedback (HM and binary diagnosis) | - .309;—.369, significant medium negative correlation at .01 level (p = < .01) |
| **Correlation between propensity to *decision switch* and *trust*** <br>*n = 216 decision points* | Decision switching perception (yes (1), maybe (2), no (3)) | Trust perception (0–5, 0 representing *no trust* and 5 *absolute trust*) following *all* AI feedback (HM and binary diagnosis) | - .007,—.009, *not* significant at .05 level (p = .902, .899) |
| **Correlation between *image quality* and agreement with *binary AI feedback*.** <br>*n = 216 decision points* | Image quality (0 –undiagnostic, 5 – excellent diagnostic quality) | Agreement with binary feedback (yes (1), partly (2), no (3)) | - .238;—.268, significant small negative correlation at .01 level (p = < .01) |
| **Correlation between *image quality* and *trust*** <br>*n = 216 decision points* | Image quality (0 –undiagnostic, 5 – excellent diagnostic quality) | Trust perception (0–5, 0 representing *no trust* and 5 *absolute trust*) following *all* AI feedback (HM and binary diagnosis | .219; .256, significant small positive correlation at .01 level (p = < .01) |
| **Correlation between *image quality* and propensity to *decision switch*** <br>*n = 216 decision points* | Image quality (0 –undiagnostic, 5 – excellent diagnostic quality) | Decision switching perception (yes (1), maybe (2), no (3)) | .086, .094, *not* significant at .05 level (p = .161, .167) |

systems, using heatmaps, is coarse, at best [34]. Other studies have found that heatmap feedback may not be as useful as hoped to clinical end-users, including radiographers, who may find provision of the performance of the system a preferred metric to improve trust [35].

Whilst the user may prefer a different form of AI feedback, this study found that there was a large, significant ($\tau_b$ = —.515, $r_s$ = —.584, $p < .01$) negative correlation between trust and extent of agreement with the heatmap, i.e., when agreement decreased (from 1 representing perfect agreement and 3 representing total disagreement), trust perception also decreased. This is important to consider in user feedback/interface design.

## 5.3 Agreement of participants with binary AI diagnosis

The participants in this study reported agreement with the AI diagnosis on 86.7% of examinations (n = 26 out of 30 examinations). Previous studies have found that users may find heatmaps confusing, leading to excessive decision switching, and binary feedback useful, leading to increased accuracy [36]. The use of expert clinicians as participants in this study allowed for the responses to be determined as reliable, and the consensus opinion as 'ground truth'. Interestingly, in cases where there was disagreement with the AI, the participants noted sub-average image quality. The established diagnosis on the images included in this study are not available (Personal communication Rajpurkar, 2019) and therefore we cannot exclude the possibly that the participants may not agree with the established diagnosis. Further study into failure analysis should be conducted.

As expected, the extent of agreement of the user with the binary feedback from the AI was correlated with trust ($\tau_b$ = —.309, $r_s$ = —.369 (moderate), $p = < .01$) i.e. the greater the extent of agreement, the greater the perception of trust.

The binary diagnosis was presented to the participants following provision of all images and heatmaps in an examination, as would be the case in the clinical setting. Studies have

found that the timing of the AI feedback may impact on the propensity of the user to follow advice from the system, resulting in confirmation and anchoring biases and potentially resulting in diagnostic error [36]. It has been suggested that the provision of any form of AI feedback should be optional, much in the same way as human-to-human feedback, however users should be aware of other biases with this method, such as confirmation bias [36]. Further research should be conducted into the timing of provision of differing forms of AI feedback on user trust.

## 5.4 Decision switching

Decision switching can be both positive, i.e., can cause the user to become more 'correct' or negative, causing the user to reach an incorrect conclusion. This has been highlighted as a potential issue when using technology to help make decisions, for example reliance on a spell-check function, or in medicine, on clinical decision support systems [37,38]. Studies in medicine, including cardiology and radiography have found that the level of experience of the user may impact their likelihood to follow the advice of the decision support system over their own decision, with inexperienced users more likely to follow the automated advice over their own [37,38]. The participants in this study were all 'experts' in reporting radiography, having both clinical and academic training, and were all currently practicing in the field. No statistically significant correlation was found between rates of decision switching and trust in this study. This may be due to the expert users having less propensity to change their mind from their initial decision, which is supported by the aforementioned studies in the area. The overall perception of the likelihood of decision switching as a result of the feedback from the AI was low, with only two instances across all datasets where participants indicated that they would change their mind in response to the AI feedback (1.1%). In most cases (n = 179, 83.5%) participants stated that they would not wish to change their initial diagnosis.

## 5.5 Image quality

Users' perception of the diagnostic quality of each image was obtained initially as a means of quality assurance, although this also permitted analysis of the impact of image quality on likelihood to change their mind in response to the AI feedback and trust in the AI. There was no correlation found between image quality perception and decision switching, although there was a statistically significant mild positive correlation between image quality and trust in AI, indicating that as image quality increased, the trust in the AI also increased. This may have importance in the clinical adoption of AI tools, where the user may be negatively impacted by poor image quality from a diagnostic perspective, as reported in other studies [39] and also have a reduced trust in any AI tool adopted to help in the decision-making process.

Finally, a statistically significant positive correlation was found between image quality and trust (n = 216, $\tau_b$ = .219 (small), $r_s$ = .256, p = < .01). This may indicate that when the user cannot clearly discern the area of interest highlighted by the AI that they are less likely to trust it. This may be an issue in situations where the AI proves to be more accurate in diagnosis that the human user due to degraded image quality. Future studies should compare the performance of human interpreters and AI for a variety of image resolutions. This may be particularly important when poor quality images are produced due to patient condition or sub-optimal technology used to acquire the images.

There were no statistically significant correlations between trust and propensity to change their mind from the initial decision, or between the quality of images and propensity to change their mind.

## 5.6 Implications for practice and recommendations

Participants in this study agreed more often with the binary AI feedback than the GradCAM heatmaps provided. This is supported by the correlation analyses suggesting that the users' trust was impacted by their level of agreement with the AI feedback. There was a stronger correlation found in trust levels with the heatmap feedback than the binary diagnosis. Developers should be aware of this when creating forms of explainable AI and further study should be undertaken to investigate the impact of different forms of AI feedback on users' trust, with particular cognisance of the potential issues regarding the use of current forms of explainable AI on the human end-user [36,40].

Decision switching was not found to be linked to trust, perhaps indicating that users in this study remain anchored to their initial decision, regardless of their levels of trust in the system, which is supported by very low levels of decision switching. Education in the functionality of AI systems and their use may allow expert users to leverage the benefits of the system to increase diagnostic accuracy. Again, developers should consider the optimal means of communicating the feedback from the system to the end users. This may mean provision of a range of formats which the clinician can use to calibrate their trust in the system for each individual task, therefore allowing flexibility in its use on a case-by-case basis.

This study should be repeated with a larger sample of clinical staff of varying expertise levels and students at different points in their educational careers, to assess the effect of AI feedback on decision switching and trust. This would allow a more comprehensive assessment of the training and awareness issues, based on clinical experience and provide evidence for development of useful interface design and explainability methods to develop appropriate trust amongst end users.

## 5.7 Limitations and suggestion for further research

Whilst this study recruited an adequate number of participants, as per current recommendation in the literature [27,28] further research should be conducted with a larger number of participants with a varied background in the production of formal diagnosis from radiographic image interpretation, such as radiologists and other healthcare practitioners who provide formal written reports as part of their role.

The participants in this study were recruited from the UK only, which was intentional in an attempt to ensure that their role and educational background was similar. However, this may mean that the findings from this study may be generalisable to countries with high income economies and similar education and practice requirements only. Studies by Amugongo et al., 2032 [41] and Ewuoso et al., 2023, [42] note that the African (and, by extension potentially other countries with low to middle income economies) perspective on the concept of 'trust' may differ. Both studies suggest that, in this context, trust may be based on collective, experiential-based factors. This is beyond the scope of this study but should be considered in the context of potential disparities in cultural norms related to trust in AI and other emerging technologies.

Additionally, consideration should be given to long-term follow up of this study in this, and similar populations after a period of time using AI technologies in clinical practice to investigate any change in perception over time.

## 6 Conclusion

Clinicians have proposed that a lack of trust is a significant barrier to the successful implementation of AI systems in the clinical setting [10] (Fazal et al., 2018). This study aimed to clarify some of the factors impacting expert users' trust in AI systems for diagnosis of abnormality

from radiographic images. The extent of user agreement with the AI heatmap and binary feedback has a positive impact on trust. Participants in this study did not agree with the localisation of GradCAM feedback, although on most occasions agreed with the binary feedback from the same AI system on the same images. This may indicate that users prefer binary feedback over this type of visual feedback and that disagreement with any form of AI feedback has a negative impact on trust perception. This will be important when designing the optimal user interfaces and forms of feedback for clinical use, where appropriate trust will ensure neither over- nor under-reliance.

## Supporting information

**S1 File. Study transcript–example image set from examination presented to participants.**
(DOCX)

**S2 File. Per-participant diagnosis/examination and agreement with AI diagnosis.**
(DOCX)

## Author Contributions

**Conceptualization:** Clare Rainey, Raymond Bond, Jonathan McConnell, Ciara Hughes, Sonyia McFadden.

**Data curation:** Clare Rainey, Raymond Bond, Jonathan McConnell, Ciara Hughes, Sonyia McFadden.

**Formal analysis:** Clare Rainey.

**Funding acquisition:** Clare Rainey, Raymond Bond, Jonathan McConnell, Ciara Hughes, Sonyia McFadden.

**Investigation:** Clare Rainey, Raymond Bond, Jonathan McConnell, Ciara Hughes, Sonyia McFadden.

**Methodology:** Clare Rainey, Raymond Bond, Jonathan McConnell, Ciara Hughes, Sonyia McFadden.

**Project administration:** Clare Rainey, Sonyia McFadden.

**Resources:** Clare Rainey.

**Software:** Devinder Kumar.

**Supervision:** Raymond Bond, Jonathan McConnell, Ciara Hughes, Sonyia McFadden.

**Validation:** Clare Rainey.

**Writing – original draft:** Clare Rainey, Devinder Kumar.

**Writing – review & editing:** Clare Rainey, Raymond Bond, Jonathan McConnell, Ciara Hughes, Sonyia McFadden.

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
