## [Decision Letter · Decision Letter 0]

6 May 2024

PDIG-D-24-00075

Reporting radiographers’ interaction with Artificial Intelligence – how do different forms of AI feedback impact trust and decision switching

PLOS Digital Health

Dear Dr. Rainey,

Thank you for submitting your manuscript to PLOS Digital Health. After careful consideration, we feel that it has merit but does not fully meet PLOS Digital Health's publication criteria as it currently stands. Therefore, we invite you to submit a revised version of the manuscript that addresses the points raised during the review process.

Please submit your revised manuscript within 60 days Jul 05 2024 11:59PM. If you will need more time than this to complete your revisions, please reply to this message or contact the journal office at digitalhealth@plos.org. Please include the following items when submitting your revised manuscript:

We look forward to receiving your revised manuscript.

Kind regards,

Ismini Lourentzou

Section Editor

PLOS Digital Health

Journal Requirements:

1. Please provide separate figure files in .tif or .eps format only and remove any figures embedded in your manuscript file. Please also ensure that all files are under our size limit of 10MB.

2. We do not publish any copyright or trademark symbols that usually accompany proprietary names, eg ©, ®, ™ (e.g. next to drug or reagent names). Please remove all instances of trademark/copyright symbols throughout the text, including © & ® on pages 2, 3, 8, 9 & 10.

Additional Editor Comments (if provided):

The reviewers commend the paper for its timely contribution to a pressing issue facing healthcare systems worldwide. However, they also point several limitations raised. While the study contributes valuable insights into trust dynamics, it's essential to acknowledge and address broader ethical and technical challenges. The paper's generalizability is questioned due to the small sample size, lack of diversity among participants, and scope on musculoskeletal radiographs, which may not generalize across different areas of radiology. Long-term follow-up and comparison with human experts are suggested by reviewers. We look forward to addressing these comments on the revised manuscript.

Staff editor note: If you plan to revise or make a new submission of this work, please declare any related manuscripts in preparation or under consideration, describe the relationship between these works and provide a copy of the related work with your submission as a 'Related Manuscript' file type. Related works available as a preprint or accepted for publication at any time during the consideration/processing of a manuscript must be clearly cited and discussed.

Reviewers' comments:

Reviewer's Responses to Questions

**Comments to the Author**

1. Does this manuscript meet PLOS Digital Health’s publication criteria? Is the manuscript technically sound, and do the data support the conclusions? The manuscript must describe methodologically and ethically rigorous research with conclusions that are appropriately drawn based on the data presented.

Reviewer #1: Yes

Reviewer #2: Partly

2. Has the statistical analysis been performed appropriately and rigorously?

Reviewer #1: Yes

Reviewer #2: Yes

3. Have the authors made all data underlying the findings in their manuscript fully available (please refer to the Data Availability Statement at the start of the manuscript PDF file)?

Reviewer #1: Yes

Reviewer #2: Yes

4. Is the manuscript presented in an intelligible fashion and written in standard English?

Reviewer #1: Yes

Reviewer #2: Yes

5. Review Comments to the Author

Reviewer #1: This is a timely and well-written manuscript that addresses a very important issue in healthcare delivery, not only for the UK, but quite broadly throughout the developed world where interpretive capacity of radiologists and non-radiologist interpreters of medical imaging is severely strained by burgeoning demand. The authors note that the development of AI has been viewed as a strategy for dealing with the need for substantial workflow augmentation in the NHS, coping with time-constraints and efficiency issues, as well as a strategy for error minimization. They cite the 2018 estimate that an effective AI system, if implemented effectively, could liberate nearly 1M hours of radiologist time per year. 

I was pleased with this paper overall, including its importance and rationale, clarity of the writing and presentation of the results, level of scientific rigor in study design and analysis, the quality of the analysis and discussion, the figures and tables, and the comprehensiveness of the reference list. 

Although many research papers in this space are focused on evaluating the performance of AI algorithms in comparison to human observers, it is unlikely that such AI systems will ultimately be used to completely replace human beings in image interpretation--rather, they will be used to augment human performance and increase efficiency. As such, it is critical that the human-machine interface be optimized. One barrier for this optimization is trust, of course, which was the main focus of this manuscript. The systems must be adapted to optimize user trust. This is not unlike any other engineering problem. Also, optimal trust would include avoidance of both under-trust and over-trust. Another area where significant knowledge gaps remain is the ideal means for AI to provide its output to the human operators. 

This paper reports the results of a pilot study that addresses both of those issues. The authors focused on radiographic images of the appendicular skeleton which, in the UK, are primarily interpreted by non-radiologists. Interpreting radiographers are a skilled and experienced set of operators, but have less training and (by extension) likely less confidence than subspecialized musculoskeletal radiologists. For this reason, they are an ideal test-group for evaluating the human/AI interface, as it would be expected that these workers would benefit the most from AI augmentation. In the study reported, the trust placed by a small sample of interpreting radiographers in the AI algorithm was quantified both during and after AI outputs in the form of either a binary diagnosis (no explanation) or a heat / salience map on the radiograph (revealing what aspect of the image the AI conclusion was based on). The presumption embedded in the study is that trust will be proportional to the "explainability" of the AI output. While this core hypothesis of AI trust is eminently reasonable, this is something which nonetheless needs to be rigorously assessed.

The methods used in this project are sound. A standard image set (the MURA validation set) of ~3,200 images from 783 patients was used to train the AI, with a similar sized validation set from the same source. The AI system was trained to produce a salience or heat-map, in order to help the observer understand what the AI conclusion was based on, i.e., the areas of spatial location used by the model to determine the probability of a fracture. The study was carried out via email on the Qualtrics platform. A total of 12 participants were recruited, which seems a reasonable number (power calculations were not possible due to novelty). Inter-rater agreement was established using Kappa for two or more raters, a standard statistical method. Consensus diagnosis was considered to be "ground truth" and descriptive statistics were focused on determination of the level of agreement between AI and human observers, and compared to the ground truth. Correlation between trust and level of agreement was determined for the two types of AI output, as well as with image quality.

The results showed moderate agreement between participants with kappa = 0.581. This result is typical for radiology research comparing human interpreters. There was a fairly high level of agreement between the human operators with the AI, both in the binary feedback (86.7% agreement) and with the heat maps (54.1%, combining perfect and partial agreements). The participants indicated that they might switch their own decision to that of the AI, a measure of trust, in only 1.1% of the cases. The correlation analysis showed a large negative correlation between users' trust and the level of agreement with the heatmap, p > 0.01, but there were no significant correlations between trust and the likelihood that the observer would change their decision based on AI feedback. The latter result could be a sampling error due to the small number of participants, however in their discussion the authors note that decision-switching is also a factor to be optimized, and excessive decision-switching is to be avoided as much as an inappropriate resistance to decision-switching leading to fewer such switches than are needed to optimize accuracy. 

In this study, it would appear that the heat-map output was more effective than had been reported in previous studies, which the authors cite. There was a stronger correlation of users' trust with the heatmaps. Also, users indicated trust of the system when their decision matched the binary-output of the AI, which the authors correctly point out could be the result of human confirmation bias. In their discussion the authors also discussed the potential impact of the timing of AI feedback to the human observer on the ability of the AI to change the mind of the human user, both from the standpoint of avoiding anchoring bias and also from avoiding undue influence of the algorithm. The authors correctly point out that the "decision-switching" metric may differ between highly experienced, subspecialized clinician-users and novices or persons with less training and experience. In this cohort of participants, the likelihood of decision-switching was considered low. 

The authors separately evaluated the effect of the diagnostic quality of the images, but this did not significantly effect the likelihood of a human observer to change their decision based on AI output. 

The authors conclude that decision-switching may be independent of trust, and suggested that this could reflect anchoring bias by the human observers, but it could also reflect the low sample size. The authors call for future work with increased sample size and diversity, and suggest also that future studies should incorporate strategies to tease out the effect of anchoring bias. The authors are wise to suggest that future study should also include a broader range of formats for providing output to the clinician-user of the AI, in order to explore which formats are most effective at engendering trust, as well as making the AI feedback optional. One of the most interesting aspects of this work, from a human-factors standpoint, is that the AI-generated heat maps engendered greater trust in the algorithm, and yet the binary feedback (without explanation) was more preferred by the users! This result is remarkable and deserves further exploration.

The figures are excellent overall, especially Figs 3, 4, 6 and 7; Table 1 is a little tedious but is rewarding to the patient reader. Table 2 is very clear. Radiographic images are excellent and add to the readers' understanding.

Overall, as noted above, I found the paper to be solid in all relevant categories. I believe that it will generate substantial reader interest, stimulate future research, and should be well cited in the years ahead. The manuscript is well-composed and written, and so have no specific suggestions for changes or improvements. I commend the author group for a solid contribution to the literature, which is both timely and needed. I hope that others will take their challenge to extend this work in the future as they have suggested.

Reviewer #2: The title of the paper is "Reporting radiographers’ interaction with Artificial Intelligence – how do different forms of AI feedback impact trust and decision switching

The paper investigates the impact of different forms of AI feedback on the trust and decision-making of reporting radiographers. 

- The results show that participants had a greater level of agreement with the binary AI feedback than with the heat map localisation. Decision switching was low, with only two instances where participants indicated they would change their initial diagnosis based on the AI feedback. 

- The study highlights the importance of developing explainable AI systems and considering the precision and accuracy of localisation to promote appropriate trust in clinical end users.

- The study highlights the importance of developing explainable AI systems and considering the precision and accuracy of localisation to promote appropriate trust in clinical end users.

However, XAI can also be biased because of bias data. A recent study has shown that clinicians can be fooled by bias AI systems, despite the system providing explanations for its outputs (jabour et al. 2023). Have the author noticed this?

Jabbour S, Fouhey D, Shepard S, et al. Measuring the Impact of AI in the Diagnosis of Hospitalized Patients: A Randomized Clinical Vignette Survey Study. JAMA. 2023;330(23):2275–2284. doi:10.1001/jama.2023.22295

Study has some limitations:

The authors claim that though they conducted their study in the UK, it can be applied globally. This may not necessarily true because trust is complex in relational communities Trust may be relational.

Lameck Mbangula Amugongo, Nicola J. Bidwell, and Caitlin C. Corrigan. 2023. Invigorating Ubuntu Ethics in A

---

## [Decision Letter · Decision Letter 1]

22 Jun 2024

Reporting radiographers’ interaction with Artificial Intelligence – how do different forms of AI feedback impact trust and decision switching

PDIG-D-24-00075R1

Dear Mrs Rainey,

We are pleased to inform you that your manuscript 'Reporting radiographers’ interaction with Artificial Intelligence – how do different forms of AI feedback impact trust and decision switching' has been provisionally accepted for publication in PLOS Digital Health.

Best regards,

Ismini Lourentzou

Section Editor

PLOS Digital Health

Reviewer Comments (if any, and for reference):

Reviewer's Responses to Questions

**Comments to the Author**

1. If the authors have adequately addressed your comments raised in a previous round of review and you feel that this manuscript is now acceptable for publication, you may indicate that here to bypass the “Comments to the Author” section, enter your conflict of interest statement in the “Confidential to Editor” section, and submit your "Accept" recommendation.

Reviewer #1: All comments have been addressed

Reviewer #3: All comments have been addressed

2. Does this manuscript meet PLOS Digital Health’s publication criteria? Is the manuscript technically sound, and do the data support the conclusions? The manuscript must describe methodologically and ethically rigorous research with conclusions that are appropriately drawn based on the data presented.

Reviewer #1: Yes

Reviewer #3: Yes

3. Has the statistical analysis been performed appropriately and rigorously?

Reviewer #1: Yes

Reviewer #3: Yes

4. Have the authors made all data underlying the findings in their manuscript fully available (please refer to the Data Availability Statement at the start of the manuscript PDF file)?

Reviewer #1: Yes

Reviewer #3: Yes

5. Is the manuscript presented in an intelligible fashion and written in standard English?

Reviewer #1: Yes

Reviewer #3: Yes

6. Review Comments to the Author

Reviewer #1: All concerns have been duly addressed. I am completely satisfied with the manuscript in its current form, and would support the publication of the companion manuscript (once it has satisfied it's own peer-review) as I believe the companion article further clarifies issues that were raised in the current review by the other reviewers.

Reviewer #3: Authors have addressed the comments in the revision and it is acceptable for publication except for one comment: section 4.2 the description of kappa statistics should be moved to the previous section on data analysis in the Methods.

7. PLOS authors have the option to publish the peer review history of their article (what does this mean?). If published, this will include your full peer review and any attached files.

**Do you want your identity to be public for this peer review?** For information about this choice, including consent withdrawal, please see our Privacy Policy.

Reviewer #1: **Yes: **Michael A. Bruno, M.D., M.S., F.A.C.R.

Reviewer #3: No
